

# Interannual variability of the ecosystem CO₂ fluxes at paludified spruce forest and ombrotrophic bog in southern taiga

Vadim Mamkin[1], Vitaly Avilov[1], Dmitry Ivanov[1], Andrey Varlagin[1], Julia Kurbatova[1]

[1]A.N. Severtsov Institute of ecology and evolution of the Russian Academy of Sciences, Moscow, Russia

*Correspondence to*: Vadim Mamkin (vadimmamkin@gmail.com)

**Abstract.** Climate warming in high latitudes impacts $CO_2$ sequestration of northern peatlands through the changes in both production and decomposition processes. The response of the net $CO_2$ fluxes between ecosystems and the atmosphere to the climate change and weather anomalies can vary across the forest and non-forest peatlands. To better understand the differences in $CO_2$ dynamics at forest and non-forest

boreal peatlands induced by changes in environmental conditions the estimates of interannual variability of the net ecosystem exchange (NEE), total ecosystem respiration (TER) and gross primary production (GPP) was obtained at two widespread peatland ecosystems – paludified spruce forest and adjacent ombrotrophic bog in the southern taiga of west Russia using 6-year of paired eddy covariance flux measurements. The period of measurements (2015-2020) was characterized by both positive and negative

annual and growing season air temperature and precipitation anomalies. Flux measurements showed that in spite of the lower growing season TER (332…339 gC·m⁻²) and GPP (442…464 gC·m⁻²) rates the bog had a lower NEE (-132…-108) than the forest excepting the warmest and the wettest year of the period and was a sink of atmospheric $CO_2$ in the selected years while the forest was a $CO_2$ sink or source between years depending on the environmental conditions. Growing season NEE at the forest site was between -

142 and 28 gC·m⁻², TER between 1135 and 1366 gC·m⁻² and GPP between 1207 and 1462 gC·m⁻². Annual NEE at the forest was between -62 and 145 gC·m⁻², TER between 1429 and 1652 gC·m⁻² and GPP between 1345 and 1566 gC·m⁻² respectively. Anomalously warm winter with sparse and thin snow cover lead to the increased GPP as well as lower NEE in early spring at forest and to the increased spring TER at the bog. Also, the shifting of the compensation point to the earlier dates at the forest and to the later dates at

the bog following the warmest winter of the period was detected. This study suggest that the warming in winter can increase $CO_2$ uptake of the paludified spruce forests of southern taiga in non-growing season.



## 1. Introduction

$CO_2$ net ecosystem exchange (NEE) between peatlands and the atmosphere is an important process of the global carbon cycle controlling the terrestrial carbon stocks and influence the climate system (Gorham, 1991; Aurela et al., 2002; Wieder and Vitt, 2006). Nothern peatlands store about $500\pm100$ Gt of C (Yu, 2012) which is approximately equals to the global vegetation and about 20-30% of the soil carbon stocks (Friedlingstein et al., 2019). Annual $CO_2$ uptake of the peatlands in high latitudes are relatively small (Moore, 2002; Koehler et al, 2010) due to the low productivity and decomposition rates limited by wet anoxic conditions, low temperatures and pH as well as low nitrogen content in the peat (Wieder and Vitt, 2006; Weedon et al., 2013). However, northern peatlands are considered to be a stable sink of atmospheric $CO_2$ at the long time scales (Gorham, 1991; Moore, 2002; Alexandrov et al., 2020) as the carbon accumulation by GPP exceeds carbon loss through $CO_2$ release from total ecosystem respiration TER and from the other mechanisms i.e. methane emissions and dissolved organic carbon (DOC) runoff.

A warming trend in high latitudes is able to intensify both carbon accumulation and release processes affecting NEE as well as net ecosystem carbon balance of the northern peatlands (Loisel et al., 2021). It is suggested that growing air and peat temperatures especially under raising frequency of droughts in boreal regions can significantly increase decomposition rates and switch peatlands from $CO_2$ sink to $CO_2$ source for the atmosphere (e.g. Alm et al., 1999; Moore, 2002; Lund et al., 2012; LaFleur et al., 2015; Loisel et al., 2021). The response of the peatlands on climate change and weather anomalies may differ across the ecosystems depending on peatland type, local weather and hydrological regime as well as vegetation composition and management practices (Humphreys et al., 2006; Euskirchen et al., 2014; Petrescu et al., 2015; Holl et al., 2019; Qiu et al., 2020). Recent decades a numerous experimental studies showed a high spatial and temporal variability of $CO_2$ fluxes between different peatland ecosystems in high latitudes and its response to environmental factors (e.g. Alm et al., 1999; Humphreys et al., 2006; Lindroth et al., 2007; Minkkinen et al., 2018; Park et al., 2021). Previous studies reported that NEE of the peatlands is susceptible to water table depth (WTD) dynamics, air and peat temperature variations,





changes in global radiation, timing of the snowmelting and peat layer thaw (Dunn et al., 2007; Lindroth et al., 2007; Sulman et al., 2010).

Forest and non-forest peatlands have a different features which determine the ecosystem – atmosphere carbon dioxide exchange: i.e. aboveground biomass, peat thickness, nutrient availability as well as different temperature and moisture regime of the upper peat layer (Moore, 2002; Kurbatova et al., 2013; Euskirchen et al., 2014; Beaulne et al., 2021). At the short time scales the forest peatlands (i.e. paludified forests) can have a similar carbon accumulation rates as the non-forest peatlands (i.e. bogs) but the forest

peatlands have a lower $CO_2$ sequestration rates at the long time scales (Beaulne et al., 2021). Moreover, NEE of the forest and non-forest peatlands have their own seasonal specifics. For instance, the forest peatlands can sequestrate atmospheric $CO_2$ before the snowmelting and peat thaw in spring, while a thawing is necessary for the beginning of the $CO_2$ uptake at non-forest peatlands (Tanja et al., 2003; Euskirchen et al., 2014). Therefore, the changes of the environmental conditions can influence the $CO_2$

fluxes at the forest and non-forest peatlands in different ways. With regard to the strong dependence of NEE on the environmental parameters variability, regional and site-specific features of the peatlands as well as its significant potential feedbacks to the climate system in response to the global warming (IPCC, 2014; Helbig et al., 2020; Loisel et al., 2021) the experimental estimates of the interannual variability of the ecosystem $CO_2$ fluxes at different peatlands located in the same landscape are very useful to assess

the diversity of the possible effects of the weather anomalies and climate change on the ecosystem carbon dioxide exchange between the northern peatlands and the atmosphere (Lavoie et al., 2005; Ueyama et al., 2014; Park et al., 2021).

Unfortunately, in spite of a numerous experimental studies focused on ecosystem-atmosphere $CO_2$ fluxes in different peatland types in high-latitudes in North America (e.g. Roulet et al., 2007; Gill et al., 2017),

Europe (e.g. Kurbatova et al., 2002; Lindroth et al., 2007; Minkkinen et al., 2018) and Asia (e.g. Tchebakova et al., 2015; Alekseychik et al., 2017; Park et al., 2021) there is lack of studies considering the ecosystem $CO_2$ fluxes at the forest and non-forest peatlands located in the same landscape and undergo the similar weather conditions (e.g. Euskirchen et al., 2014; Zagirova et al., 2019).

Russian peatlands cover about one-third of the global peatland area (Vompersky et al., 2011).

Approximately 15% of forest and non-forest peatlands in Russia are located in the European part of the





country and belongs to the most widespread ecosystems in west Russia (Vompersky et al., 2011). However, the ecosystem flux data collected at peatlands in European Russia and available in literature (e.g. Kurbatova et al., 2002; Kurbatova et al., 2008; Zagirova et al., 2019) is very sparse and rarely cover a several years of measurements, which confines the research of the dependence between $CO_2$ fluxes and

environmental conditions at interannual time scales.

This study is focused on $CO_2$ peatland-atmosphere exchange at two widespread ecosystem types in west Russia – ombrotrophic bog and paludified spruce forest located in west part of Valdai hills. The aim of the study was to analyze the interannual variability of NEE, TER and GPP at the ombrotrophic bog and paludified spruce forest located in the same landscape and to establish the response of $CO_2$ fluxes on

interannual variability of environmental conditions using 6-years of paired eddy covariance flux measurements.

## 2. Methods

### 2.1 Study sites.

This study was conducted at paludified spruce forest and adjacent ombrotrophic bog located on the

territory of the Central-Forest state natural biosphere reserve (CFSNBR) in the south-western part of Valdai hills in Tver region of Russia (Fig.1a). The sites are located 7.5 km apart and characterized by very similar weather conditions.



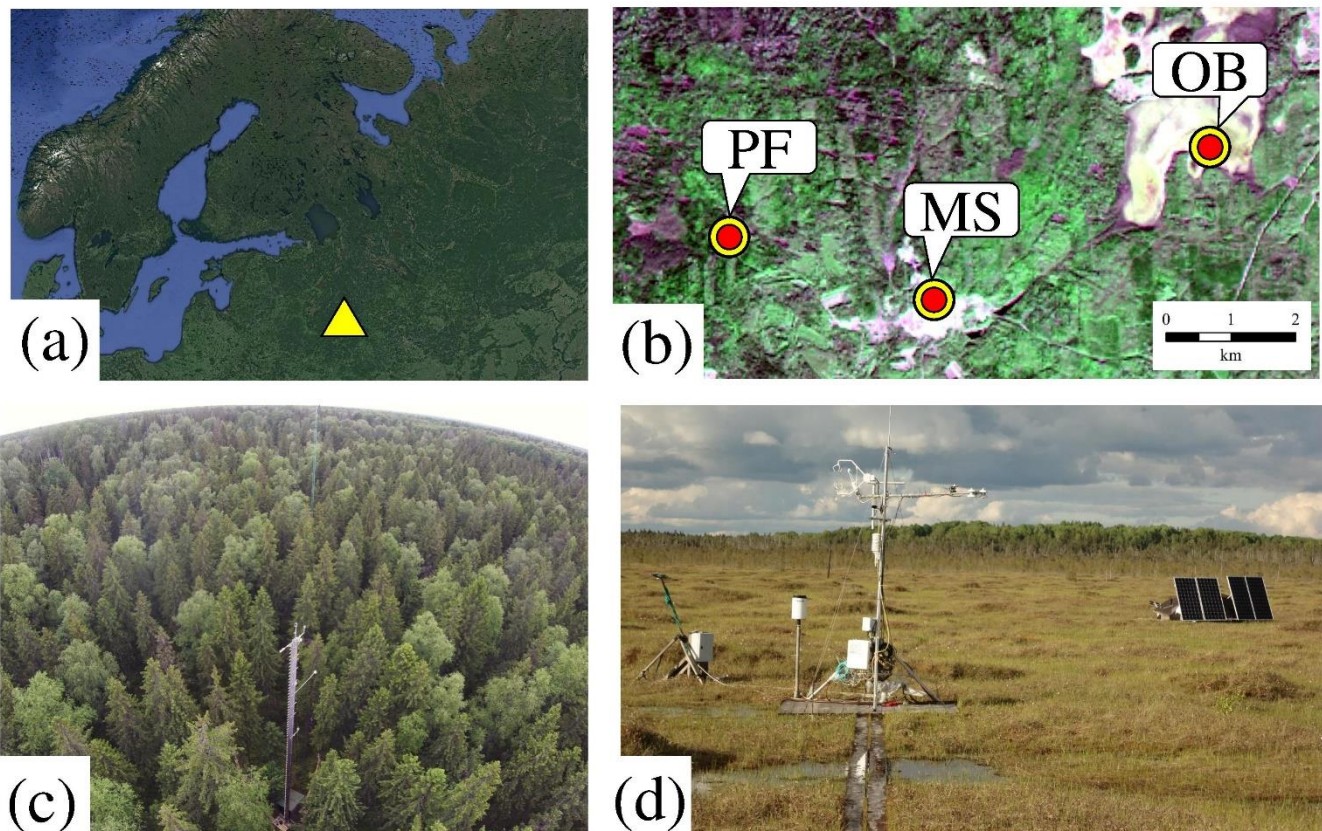

Figure 1. (a) Geographical location of the Central-Forest state natural biosphere reserve (CFSNBR) (from © Google Earth Data SIO, NOAA. U.S. Navy, NGA. GEBCO. Image IBCAO. Image LANDSAT/ Copernicus); (b) location of the paludified spruce forest (PF), ombrotrophic bog "Staroselsky Mokh" (OB) and meteorological station " Lesnoy Zapovednik" (MS) on LANDSAT-8 image; photos of the eddy covariance stations at (c) PF and (d) OB sites.

The study area belongs to the humid continental climate (Dfb type in the Köppen-Geiger climate classification) (Kuricheva et al., 2017; Peel et al., 2007). According to the meteorological station "Toropets" (56.48° N, 31.63° E, 187 m a.s.l.), which is located 80 km to west from the reserve, mean air temperature at 2 m height for the period 1991-2020 was 5.7℃ (-5.9 ℃ in January and 18.2 ℃ in July). Long-term mean annual precipitation (1991-2020) measured at meteorological station "Lesnoy Zapovednik" (56.50° N, 32.83° E, 240 m a.s.l.) – the nearest meteorological station to the study area was





778 mm. Soil surface is typically covered by snow from mid-November to late March - early April
(Desherevskaya et al., 2010) and the growing season lasts 182 days on average (since 12 Apr. to 11 Oct.).
Annual precipitation in the study region exceed potential evapotranspiration (PET), that determines
excessive moistening conditions (Mamkin et al., 2019). The climate moisture index (CMI) (Wilmott and
Feddema, 1992) ranged between 0.3 and 0.4 (Mamkin et al., 2019; Novenko et al., 2018). In the recent

30 years a significant positive trend of air temperature (+0.73 °C per10 years) and precipitation (+3.6
mm·month$^{-1}$ per 10 years) was detected in the region.

The vegetation of the reserve is represented by typical plant communities of southern taiga, which are
widespread at the plains in the north part of Eastern Europe (Mamkin et al., 2019; Shulze et al., 2002;
Vygodskaya et al., 2002). Excessive moistening coupled with spread of glacial clay soils at the territory

make the study area favourable to paludification processes and peat formation. Large areas of the
CFSNBR are covered by fens, bogs and paludified forests (Shulze et al., 2002; Puzachenko et al., 2014).
The paludified spruce forest (PF) named as RU-Fyo in FLUXNET database (56.4615°N, 32.9221°E, 265
m a.s.l.) is located in a shallow depression (Kurbatova et al., 2013) on the peaty podzolic gley soils. PF
is an old (with age up to 200 years) forest with Norway spruce (*Picea abies* - 86%) and white birch (*Betula*

*pubescens* - 14%) and undergrowth dominated by Girgensohn's sphagnum (*Sphagnum girgensohnii* L.)
and blueberry (*Vaccinium myrtillus* Russ.) (Milyukova et al., 2002; Kurbatova et al., 2008; Kuricheva et
al., 2017). Tree height reaches 27 m and undergrowth is about 0.3 m. Average leaf area index (LAI) at
PF site is 3.5. The thickness of the peat layer at the site is 60 cm with most of the root biomass at the
depth about 30 cm. The ground water table is usually close to the surface. Peat soils have a poor soil

aeration, low pH (3.5–3.8) and low nitrogen content (0.5–9.9 kg ha$^{-1}$) (Kurbatova et al., 2013; Milyukova
et al., 2002; Vygodskaya et al., 2002).

The "Staroselsky Mokh" site (OB) named as RU-Fy4 in FLUXNET database (56.4727° N, 33.0413° E,
240 m a.s.l.) is an old (up to 9000 years) ombrotrophic bog and has an area about 6 km$^2$ (Ivanov et al.,
2021). Bog's vegetation is represented by different plant communities associated with topography

130    microforms (i.a. ridges, hummocks and hollows). Vegetation of ridges and hummocks are dominated by
different herbaceous species: *Drosera rotundifolia* L., *Rhynchospora alba* (L.) Vahl*., Eriophorum*
*vaginatum* L.; shrubs: *Chamaedaphne calyculata* (L.) Moench, *Andromeda polifolia* L., *Rhododendron*



*tomentosum* Harmaja, *Vaccinium oxycoccos* L. and mosses: *Sphagnum fuscum* (Schimp.) H. Klinggr., *S. medium* Limpr., *S. angustifolium* (C.E.O. Jensen ex Russow) C.E.O. Jensen. Vegetation of hollows is dominated by herbs: *Scheuchzeria palustris* L., *Rhynchospora alba* (L.) Vahl, *Carex limosa* L., *Drosera anglica* Huds., *Eriophorum vaginatum* L. and mosses: *Sphagnum fallax* (H. Klinggr.) H. Klinggr., *S. majus* (Russow) C.E.O. Jensen, *S. cuspidatum* Ehrh. ex Hoffm., *S. balticum* (Russow) C.E.O. Jensen, *Odontoschisma fluitans* (Nees) L. Söderstr. & Váňa, *Gymnocolea inflata* (Huds.) Dumort. (Ivanov et al., 2021). Bog edges are covered by trees, predominantly by Scots pine (*Pinus sylvestris*) with tree heights up to 10-13 m. In the central part of the bog small pine trees (within 2-5 m height) grow on some ridges. Average peat layer thickness at OB site is 3.2 m with maximum of 5.5 m (Ivanov et al., 2021). Ground water level is typically 30 cm below the surface at ridges and hummocks and 10 cm above the surface at hollows.

## 2.2 Measurements

Flux stations at PF and OB sites have a standard design and instrumentation for FLUXNET network. Flux measurements at PF site started in 1998 (Kurbatova et al., 2013). Eddy covariance instruments are mounted on the top of 29 m tower located in the central part of the ecosystem (Fig. 1c). Flux measurements were obtained using 3-D sonic anemometer Gill Solent R3 (Gill Instruments, UK) and closed-path $CO_2/H_2O$ gas analyzer LI-6262-3 (LI-COR Inc., USA). Eddy covariance data was collected on the flash-drive using personal computer with EddyMeas data acquisition software (Kolle and Rebmann, 2007). Global radiation was measured using radiometer CNR4 (Kipp & Zonen B.V., Netherlands) at 28 m height. Air temperature and relative humidity measurements were carried out at 28 m height using humidity and temperature probe HMP35D (Vaisala Inc., Finland) as well as atmospheric pressure using PTB101B (Vaisala Inc., Finland) at the same height. Precipitation was measured using tipping bucket rain gauge 52202H (R. M. Young Company, USA) at 20 m height. WTD was measured using pressure transducer CS451 (Campbell Sci. inc., USA) at 1.8 m depth. Soil temperature measurements at 5 cm depth were obtained using 3 reflectometers CS650 (Campbell Scientific, USA). Meteorological data was collected every 10 s using data logger Dl 3000 (Delta-T Devices Ltd, UK).





Eddy covariance and meteorological instruments at OB site were installed in 2015 at 3.5 m tripod which
       was placed in the central part of the bog (Fig. 1d). Instruments for flux measurements included 3-D sonic
       anemometer CSAT-3 (Campbell Sci. Inc., USA) and open-path $CO_2/H_2O$ gas analyzer LI-7500A (LI-
       COR Inc., USA) mounted at 2.85 m height. Eddy covariance data was collected using LI-7550 Analyzer
       interface unit (LI-COR Inc., USA) at frequency of 10 Hz.

Additionally, global radiation was measured with 4-component radiometer NR01 (Hukseflux Thermal
       Sensors, The Netherlands) at 2.5 m height. Air temperature, relative humidity and atmospheric pressure
       was measured using weather transmitter HMP155 (Vaisala Inc., Finland) at 2 m height. Precipitation was
       measured by rain gauge Young 52202 (R. M. Young Company, USA) at 1m height near the tripod. WTD
       measurements were obtained using submersible pressure transducer CS451 (Campbell sci. Inc., USA)

installed 1.67 m below the surface. Temperature of the peat layer at 5 cm depth was measured by 3
       temperature probes T109 (Campbell sci. Inc., USA) placed in hollow, in hummock and between them.
       Meteorological data was collected every 1 min using data logger CR1000 (Campbell sci. Inc., USA). The
       Moscow time (UTC+3) was used for data storage.

**2.3 Data processing**

       This study is based on eddy covariance and meteorological data obtained at PF and OB sites in 2015-
       2016. Net ecosystem exchange (NEE) at two sites was calculated for 30-min intervals using EddyPro
       software (LI-COR Inc., USA) with all required statistical tests and corrections. Footprint was estimated
       using Kljun et al. (2004) model. 0-2 quality flags (Mauder and Foken, 2006) were assigned to calculated

fluxes. All fluxes with quality flag 2 was removed from the analysis. Additionally, all data containing the
       spikes, collected under rain and dew events as well as under low turbulence were also filtered out. Storage
       terms were calculated using one-point approach (Greco and Baldocchi, 1996) and added to $CO_2$ flux
       values. u*-filtering of NEE, gap-filling and NEE partitioning into GPP and TER was carried out using
       REddyProc package (Wutzler et al., 2018).

Mean annual u*-threshold for NEE at PF site varied between 0.354 and 0.529 m·s$^{-1}$ and between 0.058
       and 0.064 m·s$^{-1}$ at OB site.





### 2.4 Parametrization the dependence of $CO_2$ fluxes on environmental factors

The main ambient factors controlling $CO_2$ fluxes at the terrestrial ecosystems under the absence of water
stress are soil and air temperatures and the global radiation (Rg). To research how TER and GPP rates
varied following changes in environmental conditions we considered the dependence of the night-time
TER on soil and air temperature and dependence of GPP on Rg. Only original NEE data was used for
calculation TER and GPP for this analysis. To describe the dependence of TER on air and soil temperature
a widely used $Q_{10}$ function was implemented. $Q_{10}$ and $R_{10}$ coefficients were calculated following (Pavelka
et al., 2007):

$$Q_{10} = \exp(10 \cdot \alpha) \tag{1}$$

where, $\alpha$ is an empirical parameter taken from the following equation:

$$Ln(TER) = \alpha \cdot T + \gamma \tag{2}$$

Where T is soil or air temperature [°C] and $\gamma$ is an empirical parameter of the equation.

The dependence between GPP and Rg [W·m$^{-2}$] was described using well-known Michaelis - Menten
hyperbolic light-response curve:

$$GPP = \frac{\alpha \cdot \beta \cdot Rg}{\alpha \cdot Rg + \beta} \tag{3}$$

Where $\alpha$ and $\beta$ are empirical parameters of the equation. $\alpha$ is a canopy light utilization parameter [μmol·J]
and $\beta$ is a maximum $CO_2$ uptake at light saturation [μmol·m$^{-2}$·s$^{-1}$] (Matthews et al., 2017).

### 2.5 Additional data

Analysis of the weather conditions in the period 2015-2020 as well as calculation of the mean long-term
values of the meteorological parameters is based on the data collected at the two meteorological stations.
Mean air temperature data downloaded from the RIHMI-WDC database (http://aisori-m.meteo.ru)
measured at "Toropets" station was used. Only precipitation and snow cover data collected at the nearest
to the sites meteorological station "Lesnoy Zapovednik" was used considering the non-uniform spatial





distribution of the precipitation in the region and the lack of air temperature data collected at "Lesnoy Zapovednik" station.

## 3. Results

### 3.1 Meteorological conditions

Six year of measurements showed a wide interannual variability of the meteorological conditions (Fig.2). According to the data from the meteorological station "Toropets" mean annual air temperature at the period 2015-2020 was higher than the long term mean value for the period 1991-2020 (Table 1) excepting 2017 when the mean annual air temperature anomaly was not observed. Analysis of the precipitation data from meteorological station "Lesnoy Zapovednik" showed that annual precipitation in 2015 and 2018

was lower than the mean long-term annual precipitation sum and higher in 2016, 2017, 2019 and 2020. Mean air temperature calculated for the long-term growing season period (LTGS, 12 Apr. – 11 Oct.) was lower in 2015, 2017 and 2019 than the long-term mean for the same period and higher in 2016, 2018 and 2020. LTGS precipitation was lower in 2015 and 2018 and higher in 2016, 2017, 2019 and 2020. Growing season precipitation correlated with annual precipitation sums but proportion between them increased

from 45% in 2015 to 65% in 2020. The dependence between mean annual air temperature and growing season length (GSL) calculated as the number of days between the first 5-day period with mean daily air temperatures above 5°C (Leaf-on day) to the first 5-day period with mean daily air temperatures below 5°C (Leaf off day) following (Urban SIS, 2018; Buitenwerf et al, 2015; Donat et al, 2013; Mueller et al, 2015) was not established.

Therefore, the environmental conditions in the selected years were notably different. The 2017 was the coldest year of the period with the lowest global radiation and relatively high annual and growing season precipitation. In contrast, 2018 was relatively warm with highest global radiation and lowest annual and growing season precipitation. The warmest and the wettest year (as well as growing season) of the period was 2020.

All winters (Nov.-Mar.) of the selected period (2015-2020) excepting winter 2017/2018 were warmer in relation to the long-term means but the mean winter air temperatures were primarily negative (Table 2). Positive mean winter air temperature was observed in winter 2019/2020. Snow cover formed from late





October to beginning of January and was melting in April with snow depth reaching 40 cm. In winter

2019/2020 snow cover was anomalously sparse and thin 0-10 cm and was observed only since the first

week of January to mid-February.

Figure 2 Seasonal variation of mean daily air temperature ($T_a$) at meteorological station "Toropets",

10-day precipitation sums at meteorological station "Lesnoy Zapovednik" (MS) (a), daily global





radiation sums at paludified forest (PF) (b), mean daily soil temperature at 5 cm depth ($T_s$) (c) as well as inverse value of water table depth (WTD$^{-1}$) at PF and ombrotrophic bog (OB) sites correspondingly in the period 2015-2020.

Table 1. Meteorological conditions in the period 2016-2020: mean annual air temperature ($T_a$), mean air temperature calculated for the long-term growing season (LTGS, 12 Apr – 11 Oct) ($T_{a,g.s.}$) and growing season length (GSL) at meteorological station "Toropets", annual sum of precipitation (Pr) and sum of precipitation calculated for LTGS period ($Pr_{g.s.}$) at meteorological station "Lesnoy Zapovednik" (MS) as well as long-term (1991-2020) mean values of $T_a$, $T_{a,g.s}$, Pr and $Pr_{g.s}$; annual sums of global radiation (Rg), mean annual soil temperature ($T_s$) at 5 cm depth and mean annual water table depth at paludified forest (PF).

| | 2015 | 2016 | 2017 | 2018 | 2019 | 2020 | Long-Term |
|---|---|---|---|---|---|---|---|
| $T_a$ [°C] | 6.8 | 5.8 | 5.7 | 6.0 | 7.0 | 7.6 | 5.7 |
| $T_{a, g.s.}$ [°C] | 13.5 | 14.3 | 12.3 | 14.8 | 13.4 | 13.8 | 13.6 |
| GSL [days] | 180 | 187 | 174 | 194 | 172 | 178 | 182 |
| Pr [mm] | 671 | 864 | 956 | 560 | 848 | 992 | 778 |
| $Pr_{g.s.}$ [mm] | 300 | 479 | 562 | 343 | 492 | 640 | 445 |
| Rg [MJ·m$^{-2}$] | 3592 | 3402 | 3249 | 3659 | 3456 | 3333 | |
| $T_s$ [°C] | 9.0* | 6.7 | 6.0 | 6.7 | 6.6 | 6.7 | |
| WTD [m] | 0.63 | 0.28 | 0.13 | 0.37 | 0.16 | 0.16 | |

*- $T_s$ was calculated for 03.07 – 31.12 2015.

Table 2. Mean air temperature in winters (1Nov – 31 Mar) in 2015-2020 period as well as mean long-term value of air temperature at meteorological station "Toropets" [°C].

| 2015/2016 | 2016/2017 | 2017/2018 | 2018/2019 | 2019/2020 | Long-Term |
|---|---|---|---|---|---|
| -2.2 | -3.0 | -3.5 | -2.4 | 1.3 | -3.5 |





Soil temperature and water table depth dynamics at the sites were influenced by seasonal and interannual
changes of the weather conditions. Mean annual soil temperature at 5 cm depth was higher at OB site
than at PF site in growing season of the each year: mean daily soil temperature in summer reached 22.9
°C while at PF site it didn't exceed 13.5 °C. On the contrary, in winter soil temperature at PF site was
higher than at OB site: mean daily soil temperature in winter varied between 1 and 3 °C at PF site while
it reached 0 °C at OB site. WTD at the sites had a seasonal variation, so the minimal values were very
similar and observed after the snowmelting in spring of each year (-0.20 m at OB and -0.16 m at PF), then
WTD usually increased to late August and September to 1.2 m at PF site and to 0.15 m at OB site. In spite
of the difference in precipitation between years ground water at OB site was close to surface, when WTD
at PF site increased in summer to it's maximal values in the years with lower precipitation and was within
0.40-0.50 m in the relatively wet years.


### 3.2 Ecosystem $CO_2$ fluxes

Eddy covariance $CO_2$ flux measurements showed a wide seasonal and interannual variability connected
with changes in environmental conditions among the study period (2015 – 2020). During the 6 years of
measurements annual sums of NEE at PF site tended to decrease. PF was a $CO_2$ source in 2015, 2016,
2017 and 2019 and a $CO_2$ sink in 2018 and 2020 (Table 3). Annual sums of $CO_2$ at OB site were obtained
only for 2020 when OB was a stronger $CO_2$ sink than PF site, but the annual sums of GPP and TER was
lower in 3.0 - 3.5 times respectively. Mean annual GPP/TER ratio at PF site varied between 0.85 – 1.04
in the period 2015 – 2020 and was 1.23 at OB site in 2020. Strong dependence of annual NEE, TER and
GPP on GSL was not found. Minimal values of annual GPP at PF site were observed in the years with
relatively low air temperatures (2016 and 2017) and the relatively high values of annual GPP
corresponded to years with high global radiation (2015, 2018 and 2019). Annual sums of TER at PF site
were minimal in the years with maximal precipitation and correspondingly with minimal mean annual
WTD.

The annual TER and GPP were mainly determined by its growing season sums. Growing season sums
(Table 4) of TER and GPP at PF site (calculated for the long-term climatic growing season 12.04-11.10)
made up 84-86% and 90-92% respectively in 2015-2019. In 2020 due to the anomalously warm winter





2019/2020 characterized by sparse and thin snow cover growing season sums were 76% of annual TER and 86% of annual GPP. At OB site growing season sums in 2020 were 81% of annual TER and 92% of annual GPP. We hypothesize that growing season sums at OB site in the other years may exceed 95 % of

annual GPP. Comparison of the growing season NEE for 2016, 2019 and 2020 at two sites showed that OB site was a $CO_2$ sink in all selected years while PF site was a $CO_2$ source in 2016, moreover NEE at OB site was lower than at PF site in spite of the lower GPP rates in 2016 and 2019 but in 2020 lower growing season NEE at PF site was detected (Table 4) The GPP/TER ratio in growing season was 0.98 – 1.12 at PF site and 1.32 – 1.40 at OB site.

Similarly, the lowest winter sums (01 Nov – 31 Mar) of NEE at PF site were detected in relatively warm years that is mostly connected with increased GPP. Winter GPP in the warmest winter was higher than in the coldest one on 65% (43 $gC·m^{-2}$ in winter 2017/2018 and 123 $gC·m^{-2}$ in winter 2019/2020) while TER increased on 28% (149 $gC·m^{-2}$ in winter 2017/2018 and 206 $gC·m^{-2}$ in winter 2019/2020). At OB site all main components of NEE (TER and GPP) were lower than at PF site in winter. We compared the winter

sums of carbon dioxide fluxes at PF and OB sites for two winter seasons: winter 2015/2016 with thick and continuous snow cover and anomalously warm winter 2019/2020 with thin and sparse snow cover. It was obtained that NEE at OB site in winter 2019/2020 was slightly higher (40 $gC·m^{-2}$) than in winter 2015/2016 (34 $gC·m^{-2}$) while NEE at PF site in winter 2019/2020 (83 $gC·m^{-2}$) was lower than in winter 2015/2016 (115 $gC·m^{-2}$). The response of $CO_2$ exchange on anomalously warm weather conditions in

winter 2019/2020 at PF and OB sites was different. At PF site GPP/TER ratio increased from 0.38 in winter 2015/2016 to 0.60 in winter 2019/2020 but at OB site it slightly decreased from 0.38 to 0.37. At PF site GPP and TER were higher in winter 2019/2020 on 42 and 9% respectively and at OB site GPP increased on 8% and TER on 12%. Therefore, warm winter lead to significant increasing of GPP at the paludified forest and small changes of GPP at the bog as well as TER at both sites.


Table 3 Annual sums of the net ecosystem exchange (NEE), total ecosystem respiration (TER) and gross primary production (GPP) and GPP/TER ratio at the paludified forest (PF) in the period 2015 – 2020 and the ombrotrophic bog (OB) in 2020.



| | **2015** | **2016** | **2017** | **2018** | **2019** | **2020** | **2020 (OB)** |
|---|---|---|---|---|---|---|---|
| **NEE [gC·m⁻²]** | 70 | 145 | 21 | -30 | 39 | -62 | -95 |
| **TER [gC·m⁻²]** | 1636 | 1652 | 1366 | 1537 | 1631 | 1429 | 410 |
| **GPP [gC·m⁻²]** | 1566 | 1408 | 1345 | 1566 | 1592 | 1491 | 505 |
| **GPP/TER** | 0.96 | 0.85 | 0.99 | 1.02 | s0.98 | 1.04 | 1.23 |


Table 4 Growing season (calculated for long-term mean growing season 12.04 – 11.10) sums of net ecosystem exchange (NEE), total ecosystem respiration (TER), gross primary production (GPP) and GPP/TER ratio at the paludified forest (PF) and at the ombrotrophic bog (OB) in the period 2015 – 2020.

| | **2015** | **2016** | | **2017** | **2018** | **2019** | | **2020** | |
|---|---|---|---|---|---|---|---|---|---|
| | **PF** | **PF** | **OB** | **PF** | **PF** | **PF** | **OB** | **PF** | **OB** |
| **NEE [gC·m⁻²]** | -43 | 28 | -108 | -72 | -135 | -78 | -120 | -142 | -132 |
| **TER [gC·m⁻²]** | 1366 | 1317 | 334 | 1135 | 1308 | 1384 | 339 | 1140 | 332 |
| **GPP [gC·m⁻²]** | 1409 | 1289 | 442 | 1207 | 1443 | 1462 | 458 | 1282 | 464 |
| **GPP/TER** | 1.03 | 0.98 | 1.32 | 1.06 | 1.10 | 1.06 | 1.35 | 1.12 | 1.40 |

Lower GPP and TER rates at OB site in comparing with GPP and TER at PF site were also detected in seasonal variability in all years of measurements (Fig. 3). Maximal daily sums of TER and GPP were observed in summer: TER reached 19 gC·m⁻²·d⁻¹ and GPP 18 gC·m⁻²·d⁻¹ at PF site, while at OB site TER didn't exceed 6 gC·m⁻²·d⁻¹ and GPP 7 gC·m⁻²·d⁻¹ respectively. As a result, seasonal amplitude of NEE at PF site was more pronounced than at OB site and the daily sums of NEE ranged between 7 and -7 gC·m⁻²·d⁻¹ at PF site and between 1 and -3 gC·m⁻²·d⁻¹ at OB site.



Figure 3 Seasonal variation of the net ecosystem exchange (NEE) (a), total ecosystem respiration (TER) (b) and gross primary production (GPP) (c) at paludified forest (PF) and ombrotrophic bog (OB) sites in the period 2015-2020.

In 2015-2019 period PF became a sink of atmospheric $CO_2$ in March (3-7 weeks before the Leaf-on day calculated using mean daily air temperature data) and $CO_2$ source in late September – mid October. OB





site became a sink after snow melting in late April (2-4 weeks after Leaf-on day) – first decade of May
and a $CO_2$ source in September. In the warmest year - 2020 at PF site the first days with daily NEE<0
were observed in mid-February (10 weeks before Leaf-on day) while at OB site only in the end of May
(5 weeks after Leaf-on day). The shift of compensation point due to the positive temperature anomaly
and lack of snow cover in winter and early spring to the earlier dates at PF site and later dates at OB site
can be explained by the difference in vegetation composition and its phenology. Primary production of
the conifer trees at PF site in February and March is limited by low air temperatures and the positive
temperature anomaly triggered early $CO_2$ uptake at the forest site thus the GPP at PF site grew faster than
TER. At OB site the lack of snow lead to the fast heating of the upper peat layer and consequently TER
increased faster than GPP. In spite of the lower NEE at OB site, time period when the PF site was a sink
of atmospheric $CO_2$ was longer primarily in spring. Therefore, the weather conditions in spring play a
key role in differences between NEE at paludified spruce forest and ombrotrophic bog.

**3.3 Environmental controls of $CO_2$ fluxes**

The main components of NEE (TER and GPP) varied during the period (2015-2020) following the
changes in the different environmental factors. The dependence between 30-min and daily sums of TER
and WTD was not found but the TER rates at the sites were sensitive to the soil and air temperatures. We
used $Q_{10}$ function (Eq.1) for parametrization the dependence of TER on air and soil temperatures at PF
and OB sites (Fig. 4). Only original night-time data of NEE collected in 12 Apr. – 11 Oct. period was
used for the analysis. Mean night-time soil temperature at OB site varied in the wider range than at PF
site from 3 °C and reaching 25 °C, while at PF site it was between 0 and 15 °C. Air temperature variations
at night were very similar at two sites. TER rates in the presented soil and air temperature ranges had a
different magnitude at the sites. TER at PF site reached 16 $\mu mol \cdot m^{-2} \cdot s^{-1}$ while TER at OB site didn't
exceed 6 $\mu mol \cdot m^{-2} \cdot s^{-1}$, moreover TER at PF site was on average higher than at OB site within the whole
presented air and soil temperature ranges.

Maximal $Q_{10}$ values at PF and OB sites were observed when soil temperature was used for $Q_{10}$ coefficient
calculation, but $Q_{10}$ values was higher at PF site than at OB site if $Q_{10}$ is calculated using soil temperature
and higher at OB site if air temperature is used for calculation (Table 5). Similarly, maximal $R_{10}$




coefficient at PF site was obtained using soil temperature and using air temperature at OB site. Regardless

of soil or air temperature was used, $R_{10}$ at PF site was significantly higher than at OB site.

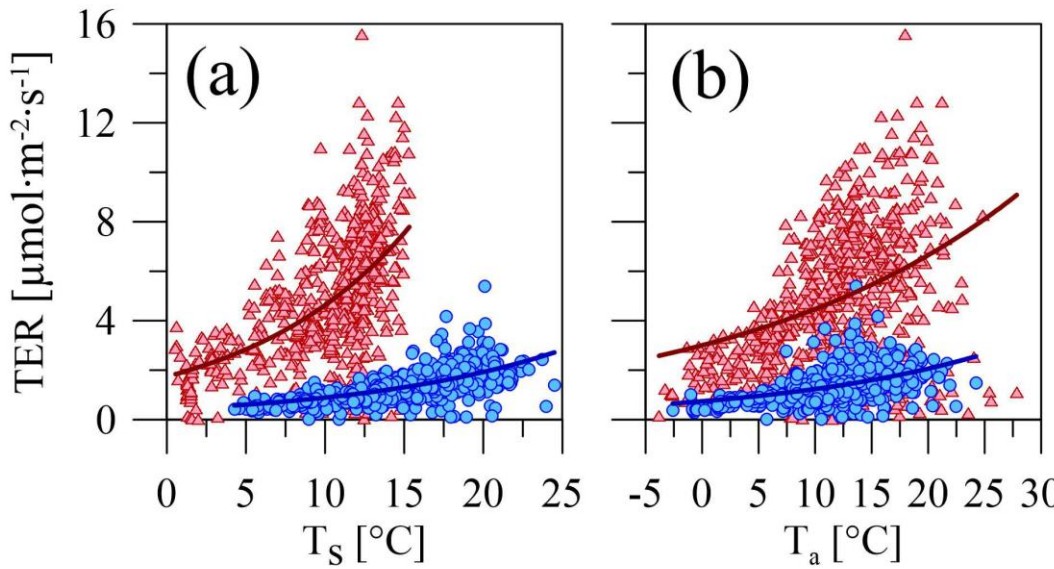

Figure 4 Relationship between the mean night-time ecosystem respiration (TER) and (a) soil ($T_s$) and
(b) air temperature ($T_a$) at paludified forest (PF) and ombrotrophic bog (OB) sites in the period 12 Apr.
– 11 Oct. 2015 – 2020 approximated using $Q_{10}$ function (Eq. 1).

Table 5. Parameters $\alpha$ and $\gamma$ of the Eq. (2) and $R^2$ ($p < 0.01$) as well as $Q_{10}$ and $R_{10}$ coefficients calculated
using soil ($T_s$) and air ($T_a$) temperature at paludified forest (PF) and ombrotrophic bog sites (OB).

|  | $\alpha$ | $\gamma$ | $R^2$ | $Q_{10}$ | $R_{10}$ [$\mu mol \cdot m^{-2} \cdot s^{-1}$] |
|---|---|---|---|---|---|
| **PF ($T_s$)** | 0.098 | 0.556 | 0.289 | 2.66 | 4.65 |





| | | | | | |
|---|---|---|---|---|---|
| **OB($T_s$)** | 0.077 | -0.883 | 0.431 | 2.16 | 0.89 |
| **PF($T_a$)** | 0.040 | 1.100 | 0.174 | 1.49 | 4.48 |
| **OB($T_a$)** | 0.051 | -0.297 | 0.261 | 1.67 | 1.24 |


To represent the dependence of GPP at the sites to changes in Rg we used the hyperbolic light-response curve (Eq. 3). Only 30-min GPP values calculated from original NEE data was taken for the analysis. To show the seasonal and interannual variations of the curve parameters we considered the data in different months (April, July and October) for anomalously cool and wet growing season 2017 and anomalously

warm and dry growing season 2018 (Fig. 5).

April is a beginning of the growing season and characterized by wide range of Rg and comparatively narrow range of GPP. Unfortunately, lack of the original NEE data at OB site in April didn't allow us to compare light-response curves parameters at the two sites at this month. July is the middle of the growing season, when Rg and GPP are relatively high. In the end of the growing season (October) both GPP and

Rg had a relatively narrow range of variation. It is important to note that GPP at OB site was lower than at PF site in all months of the selected years that largely determined the difference in the curves' parameters between the sites.

Analysis of the light-response curves (Table 6) showed that in relatively warm April 2018 α coefficient, (which refers to sensitivity of GPP to Rg) was lower than in relatively cold April 2017, but β coefficient

(which denotes the saturation point of the curve) was higher in April 2018. In July α and β coefficients at PF site were higher in the relatively warm 2018 and in the relatively cool 2017 at OB site. In July 2017 α at PF site was lower than at OB site, but β was relatively higher. In July 2018 both α and β parameters were higher at PF site. In October 2017 and 2018 α and β were also higher at PF site and the highest values of α at PF and OB sites were detected in October 2017 while highest β in October 2018.

Additionally we compared the light response curves for March (2018 and 2020) – the last winter months in anomalously cold and snowy winter 2017/2018 and anomalously warm winter 2019/2020 with sparse and thin snow cover at PF site (Fig. 6). Due to the low GPP values it was quite difficult to establish the dependence between GPP and Rg in March 2018 while the sensitivity of GPP to changes in Rg was pronounced in March 2020 with relatively high GPP in the whole range of Rg, especially at high Rg





values. It demonstrate the difference in response of GPP to interannual changes in environmental

conditions at the paludified forest in early spring.

Figure 5. Hyperbolic light response curves (Eq. 3) of gross primary production (GPP) for April (a, d), July (b, e) and October (c, f) in 2017 and 2018 at paludified forest (PF) and ombrotrophic bog (OB) sites.





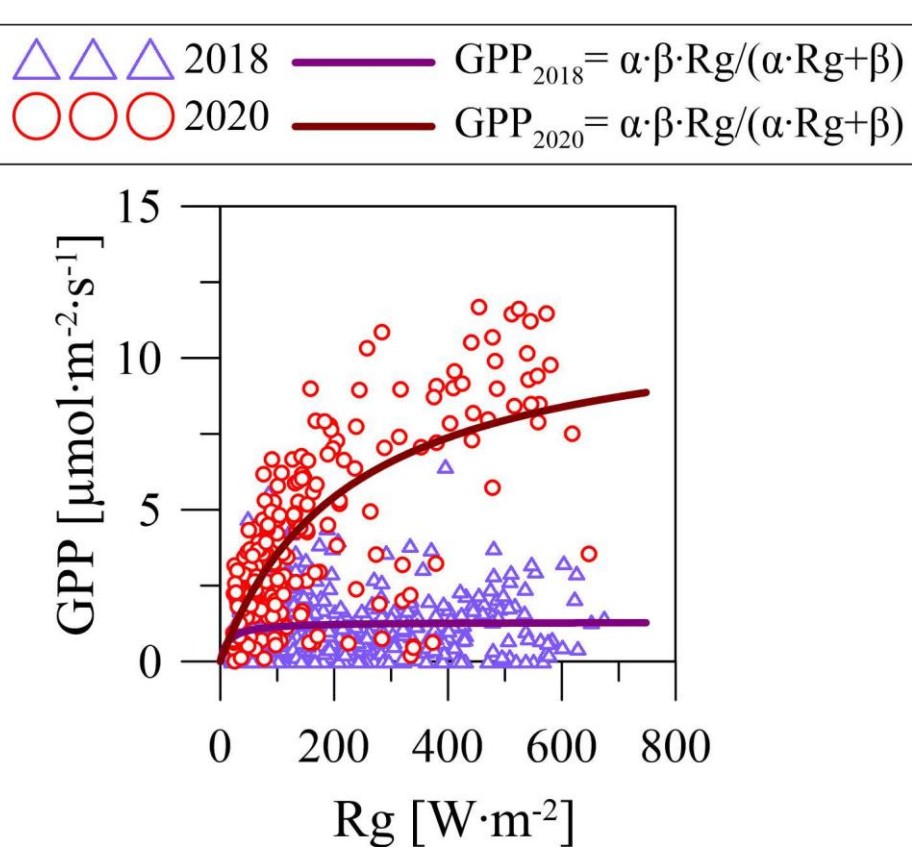

Figure 6. Hyperbolic light response curves (Eq. 3) of gross primary production (GPP) for March 2018 and 2020 at the paludified forest (PF).

Table 6. Parameters $\alpha$ and $\beta$ of the light response curves (Eq. 3) and $R^2$ (p<0.01) for March, April, July and October in 2017, 2018 and 2020 at the paludified forest (PF) and the ombrotrophic bog (OB).

| | $\alpha$ [$\mu mol \cdot J^{-1}$] | | $\beta$ [$\mu mol \cdot m^{-2} \cdot s^{-1}$] | | $R^2$ | |
|---|---|---|---|---|---|---|
| | **PF** | **OB** | **PF** | **OB** | **PF** | **OB** |
| **2017** | | | | | | |
| **April** | 0.085 | - | 8.3 | - | 0.299 | - |
| **July** | 0.112 | 0.132 | 28.9 | 12.1 | 0.567 | 0.562 |
| **October** | 0.104 | 0.048 | 18.2 | 2.4 | 0.542 | 0.134 |
| **2018** | | | | | | |



| | | | | | | |
|---|---|---|---|---|---|---|
| **March** | 0.093 | - | 1.3 | - | 0.014 | - |
| **April** | 0.067 | - | 15.5 | - | 0.403 | - |
| **July** | 0.114 | 0.062 | 35.7 | 7.2 | 0.604 | 0.585 |
| **October** | 0.101 | 0.032 | 19.3 | 4.2 | 0.610 | 0.014 |
| **2020** | | | | | | |
| **March** | 0.05 | - | 11.5 | - | 0.485 | - |

## 4. Discussion

### 4.1 Ecosystem CO₂ fluxes

Six years of paired eddy covariance measurements (2015-2020) at PF and OB sites showed that

ombrotrophic bog in southern taiga of West Russia was a stronger $CO_2$ sink in the growing season than the adjacent paludified spruce forest excepting the warmest year of the period, however paludified spruce forest had a greater annual, growing season and daily TER and GPP than the adjacent bog. Therefore under the described environmental conditions the difference in annual and growing season $CO_2$ uptake of the ecosystems was dependent mostly on specific GPP/TER ratio rather than on difference in GPP. The

growing season GPP/TER ratio at PF site was 0.98-1.12 and 1.32-1.40 at OB site and the annual value changed between 0.85 and 1.04 at PF site and was 1.23 at OB site in 2020. Annual and growing season values of GPP/TER ratio at PF site corresponded to the values obtained in old spruce forest on mineral soils located in 2 km to south from PF site (0.98) (Mamkin et al., 2019) and for black spruce (*Picea mariana*) stands on peaty soils reported by Dunn et al., (2007) in Manitoba (Canada) (annual: 0.89-1.09);

Ueyama et al. (2014) (annual: 0.85-1.41 and the growing season: 0.90-1.61) and Euskirchen et al. (2014) (annual was about 1.04 and the growing season 1.14-1.47 with annual GPP/TER ratio for adjacent bog site about 1.08 and 1.30-1.42 in the growing season) on permafrost in Alaska (USA). GPP/TER ratio for OB site is also in agreement with the growing season values reported by Sulman et al. (2010) for bogs in Wisconsin (USA) (1.03) and Ontario (Canada) (1.30) but it was lower than GPP/TER ratio for bog in

West Siberia (2.29) obtained by Alekseychik et al. (2017) in May-August period.





In spite of a numerous experiments focused on $CO_2$ fluxes at boreal and temperate peatlands (e.g. Martikainen et al., 1995; Lafleur et al., 2001; Aurela et al., 2002; Lindroth et al., 2007; Petrescu et al., 2015) it is a small number of studies based on simultaneous flux measurements at spruce forests and bogs located in the same landscape and undergo the similar weather conditions. The estimates of NEE at OB

and PF sites (Table 3 and Table 4) are similar to the annual NEE obtained by Euskirchen et al., (2014) at permafrost in Alaska (USA) during the 3 years of measurements (between -76 and 72 $gC·m^{-2}·yr^{-1}$ at black spruce forest and between -81 and 23 $gC·m^{-2}·yr^{-1}$ at the bog) as well as TER and GPP sums at OB site (465-519 $gC·m^{-2}·yr^{-1}$ and 496-548 $gC·m^{-2}·yr^{-1}$ for TER and GPP respectively) while annual TER and GPP at black spruce forest was significantly lower than at PF site (491-633 $gC·m^{-2}·yr^{-1}$ and 544-588 $gC·m^{-2}·yr^{-1}$

$^1$ for TER and GPP respectively). The similar annual and growing season NEE and larger TER and GPP sums at PF site in comparing with data from other black spruce stands was also detected. For example Ueyama et al., (2014) reported that growing season (April – September) sums of TER and GPP at the black spruce stand on permafrost in Alaska varied between 403-759 $gC·m^{-2}$ and 491-799 $gC·m^{-2}$ respectively and corresponding NEE values changed from -93 to 15 $gC·m^{-2}$ during the 9 years of

measurements, so the black spruce stand was primarily a sink of atmospheric $CO_2$. Dunn et al. (2007) also examined $CO_2$ fluxes in old black spruce forest in Manitoba (Canada) during the 9 years and found that annual NEE was -8 and 54 $gC·m^{-2}$ with annual TER 611-826 $gC·m^{-2}·yr^{-1}$ and GPP 610-782 $gC·m^{-2}·yr^{-1}$. NEE as well as TER and GPP at PF site was, also similar to the estimates for the adjacent old spruce forest on mineral soils: NEE from April to mid-October 2016 was 24 $gC·m^{-2}$ with corresponding TER

=1373 $gC·m^{-2}$ and GPP =1349 $gC·m^{-2}$ (Mamkin et al., 2019). The relatively warm and wet weather conditions with long growing season and nutrient availability in southern taiga of Valdai Hills provide a favorable conditions for photosynthesis and ecosystem respiration of the boreal forests (Karpov, 1983) but its annual $CO_2$ uptake under the environmental conditions close to the long-term means is similar to the estimates from the other boreal forests located in colder climate and moreover is lower than the annual

uptake at the bogs of the same landscape.

Larger $CO_2$ uptake at the Siberian spruce (*Picea obovata* Ledeb.) forest on peaty soils than at mesotrophic peatland located in 40 km from the forest in Nothern Ural (Komi republic, Russia) in April-August period was measured by Zagirova et al. (2019): NEE at the forest was -327 $gC·m^{-2}$ and -140 $gC·m^{-2}$ at the





mesotrophic peatland. NEE as well as TER and GPP sums for OB site is also corresponded to the
estimates from other studies: For example NEE of the bog in Western Siberia in the period May-August
as estimated by Alekseychik et al. (2017) was -202 gC·m$^{-2}$ with corresponded TER and GPP values 157
gC·m$^{-2}$ and 359 gC·m$^{-2}$ respectively. Annual NEE at the bog in southern Sweden reported by Lund et al.
(2007) was -21.5 gC·m$^{-2}$.

Interannual variability and the long-term trends in environmental conditions could substantially influence
annual NEE at southern taiga peatlands. The NEE estimated at PF site in 2015-2020 period (Table 3) was
lower than NEE reported by Kurbatova et al. (2008) and measured at the same site in 1999 – 2004 period
(100-600 gC·m$^{-2}$·yr$^{-1}$). The difference could be explained by positive air temperature trend in the last 20
years that enhanced GPP at the site and increasing in annual precipitation which therefore provided
decreasing in WTD and consequently inhibited TER. The decreasing of NEE at PF site due to the lowering
in WTD is in agreement with model experiments carried out by Kurbatova et al. (2008).

Interannual variation of the growing season NEE at PF and OB sites as well as interannual variation of
annual NEE in 2015-2020 is explained by changes in both TER and GPP rates and the maximal GPP
values were observed in the years with increased Rg. Previous study at PF and OB sites (Kurbatova et al.,
2013) demonstrated that TER play a key role in the NEE of the sites, especially in the relatively warm
years. It is correspondent with (Dunn et al., 2007; Ueyama et al., 2014; Euskirchen et al., 2014) who
showed that interannual changes in annual NEE of the black spruce stands in Alaska (USA) and Manitoba
(Canada) was primarily dependent on TER variability than on changes in GPP. Strong dependence of
NEE on TER was also obtained for fen in Alberta (Canada) by Cai et al. (2010).

It is considered, that relatively warm and dry weather conditions usually increase TER rates at peatlands
due to the enhanced soil and air temperature as well as WTD and can shift a peatland from $CO_2$ sink to
$CO_2$ source (Moore, 2002; Drösler et al., 2008; Minkinnen et al., 2018). Thus, the drought events can be
the major environmental factor of interannual variations of NEE at the peatlands controlling the TER
rates (Welp et al.,2007; Lund et al., 2012). It is consistent with chamber measurements at OB site
performed in the previous years and including an extreme drought 2010 in West Russia (Ivanov et al.,
2017; Kurbatova et al., 2013). It is important to note that respiration rates measured by chambers and
reported by Kurbatova et al. (2013) reached the maximum values and were more sensitive to temperature





variations under the weather conditions close to the long-term means in summer. Under the low WTD in the snowmelting period of spring and in extreme drought 2010 (with WTD>35 cm) respiration rates were lower and less sensitive to the temperature variations. In the period 2015-2020 there wasn't such extreme
droughts and relatively low WTD variation was observed at OB site. At PF site a substantial increasing of WTD was detected in the growing seasons with the negative precipitation anomaly. In spite of the increased annual and growing season TER were detected in the years with high WTD the dependence of the 30-min and mean night-time TER on WTD was not found as well as at OB site. Therefore, the air and soil temperature was the main predictor of seasonal variations of TER at the sites in the selected years.
Weak dependence of TER and NEE to WTD at the peatlands was reported in many studies (e.g. Lafleur et al., 2001; Lafleur et al., 2005; Parmentier et al., 2009; Sulman et al., 2009; Alekseychik et al., 2017). For example, Parmentier et al. (2009) hypothesized that increasing in WTD will affect TER on peatland if WTD changes are accompanied by changes in soil water content. Lafleur et al. (2005) suggested that dependence of TER on WTD is a result of complex interactions between WTD, vertical profiles of peat
water content in the unsaturated zone above the water table, and vertical profiles of peat decomposability and supposed that TER at wetter peatlands should be more sensitive to changes in WTD. Sulman et al., (2009) detected that lowering in water level increase both TER and GPP which lead to small dependence of NEE on the peatland to changes in WTD. It is feasible that small changes in WTD and uniform distribution of precipitation during the study period preserved the peat layer properties at PF and OB sites
and the strong dependence of TER and NEE on WTD would be detectable if droughts would be more frequent in the region.

The sensitivity parameters of TER ($Q_{10}$) to soil and temperature (Table 5 and Table 6) corresponded to values obtained in other studies (Lafleur et al., 2005; Humphreys et al., 2006; Lindroth et al., 2007; Lund et al., 2007; Ueyama et al., 2014). For example Humphreys et al. (2006) obtained mid-summer $Q_{10}$ values
calculated using air temperature for bog and different fens in Canada between $1.3 - 2.0$ and $R_{10}$ was $0.9 - 3.2$ µmol·m$^{-2}$·s$^{-1}$. Lund et al. (2007) reported that $Q_{10}$ at the temperate bog in Sweeden was 1.81 when air temperature was used for calculation and 2.83 was obtained using soil temperature at 5 cm depth. Also a slightly different $Q_{10}$ were obtained depending on microtopography of the bog: corresponding $Q_{10}$ values for hummocks was 2.32 and 2.54 for hollows. Lafleur et al. (2005) estimated $Q_{10}$ values at the bog





in Ontario (Canada): 2.24 when air temperature at 50 cm height was used, 2.57 at hummock and 3.91 at hollow. $Q_{10}$ at the black spruce stand in Alaska (USA) in the growing season reported by Ueyama et al. (2014) and calculated using air temperature ranged between 1.5-2.5. However $Q_{10}$ values calculated for PF site were lower than for adjacent spruce forest on mineral soils in the growing season 2016: 2.49 and 5.77 calculated using air and soil temperature respectively and reported by Mamkin et al. (2019). The corresponded $R_{10}$ values were 5.43 and 5.77 respectively. Lower sensitivity of TER on PF and OB sites in comparing with the adjacent forest on mineral soils could be connected with both decreased heterotrophic and autotrophic respiration due to the paludification which inhibits decomposition processes in the upper soil layers and limit productivity of the ecosystems.

**4.2 Implications for climate change in the region.**

Observed warming trend in the boreal ecozone lead to the increasing in both TER and GPP (Aurela et al., 2004; Minkkinen et al., 2018; IPCC, 2019). But the net effect on NEE (shifting ecosystem status to $CO_2$ source or $CO_2$ sink for the atmosphere) can vary across the ecosystems depending on local environmental conditions, hydrology and vegetation type. The six years of eddy covariance measurements at PF and OB sites showed that positive temperature anomaly can increase proportion of the winter fluxes to the annual sums. At PF site positive anomaly in winter months mainly increase GPP while at OB site increasing of TER was more pronounced. Winter NEE in boreal peatlands is usually depended on TER rates (Fahnestock et al., 1999; Koehler et al., 2010; Lohila et al., 2011). Thus the slight growth of air and soil temperature can potentially increase $CO_2$ release. Several studies showed that winter emission in boreal peatlands can offset the summer $CO_2$ uptake (Alm et al., 1999; Lafleur et al., 2001; D'Acunha et al., 2019).

Previous studies carried out at the same sites using eddy covariance and chamber measurements as well as modelling experiments (Miliukova et al., 2002; Kurbatova et al., 2008; Kurbatova et al., 2013) suggested that climate warming can increase TER and consequently NEE of the bogs and paludified forests in the region. In this study we estimated that interannual variability of GPP driven by temperature anomaly can substantially influence annual and seasonal NEE of the selected ecosystems and positive temperature anomaly lead to the decreasing in NEE of the paludified forest. It is corresponded with Dunn





et al. (2007) who showed that black spruce forest on peaty soils in Manitoba (Canada) switched from $CO_2$ source to $CO_2$ sink in several years following the positive temperature trend. The different response of NEE on positive temperature anomaly in previous studies and in the present research is likely connected with difference in moisture regime at PF and OB sites between 1999-2011 and 2015-2020 periods. So under the hot and dry conditions ecosystems were a $CO_2$ source for the atmosphere that is in agreement with several studies at the similar ecosystems (Lund et al., 2007; Cai et al., 2010). It is likely that increasing in precipitation during the last decades which was uniformly distributed over a growing season period provided a low WTD and created a favourable conditions for growing in GPP and decreasing of NEE at the sites.

Influence of present warming trend on NEE of the peatlands is also dependent on season when the substantial changes in air temperature and precipitation are observed. Shifting of the growing season start and snowmelting period can additionally impact the annual NEE. For example, late snowmelting and the late leaf on followed by a hot and dry summer in Alaska (USA) switched black spruce permafrost forest and adjacent bog from $CO_2$ sink (under temperature and precipitation close to the long-term means) to the $CO_2$ source for the atmosphere (Euskirchen et al., 2014). Conversely, early leaf on and snowmelting can lead to the decreasing in annual NEE and therefore the environmental conditions in spring can determine the annual $CO_2$ balance of the forest and bog ecosystems that was reported in many experimental studies (Lafleur et al., 2001; Aurela et al., 2004; Syed et al., 2006; Black et al., 2000; Hommeltenberg et al., 2014). However, the research provided by Goulden et al. (1998) showed that a positive air temperature trend in spring can lead to the substantial carbon loss in an old black spruce forest in Manitoba (Canada). An important factor of the spring NEE can be a thaw water supply. For example Hu et al. (2010) reported that early seasonal warming with lack of thaw water in spring increased NEE in subalpine pine-aspen forest in Colorado Rocky Mountains (USA).

The response of $CO_2$ fluxes at PF and OB sites on positive temperature anomaly in spring was different. $CO_2$ uptake at paludified spruce forest began before the snowmelting and the anomalously warm conditions in late winter and early spring provided a substantial increasing in GPP while the $CO_2$ uptake at the bog observed after the start of the growing season. Thus we expect that increased frequency of thaw weather periods in winter that was predicted in West Russia (Roshydromet, 2014; IPCC, 2014) and early





spring can decrease NEE of the paludified forest with less significant changes of NEE at the bog. Unlike warm springs, warm autumn can increase TER more than GPP and consequently reduce an annual $CO_2$ sequestration (Piao et al., 2008; Ueyama et al., 2014).

According to the meteorological observations during the last 30 years mean annual air temperature and annual precipitation on Valdai hills show a positive trend in the last decades which is mostly connected with increasing in winter temperature and precipitation however the thickness of snowpack and the period with snow cover is decreasing. Moreover, the growing season became longer primarily due to the shifting of leaf on day to early dates in spring with no significant shift of leaf off day in autumn. According to the current climate changes an increasing of $CO_2$ sequestration at peatlands due to the enhanced GPP rates, especially at forest ecosystems, in early spring in the west part of Valdai Hills is expectable. However, the latest climate predictions (IPCC, 2014) for the region showed that future warming in the next decades will be attended with increasing in winter precipitation and decreasing in summer. Thus in spite of the positive trend in annual precipitation a raising frequency of heat waves and droughts in summer is also presumable. While warming and moistening in winter could increase GPP more than TER especially at paludified forests, the extreme hot and dry conditions are able to increase TER significantly and switch peatlands from $CO_2$ sink to the consistent $CO_2$ source for the atmosphere. Noting that: PF site is charachterized by high interannual variability of NEE which is close to 0 and a relatively high daily, growing season and annual TER and GPP as well as high sensitivity of TER and GPP to the changes in environmental factors, the observed warming trend can affect the status of the paludified forests in southern taiga as a source or sink of atmospheric $CO_2$ more than bogs located in the same landscape. Therefore, we can expect an increasing role of the moistening conditions under future climate change for the ecosystem status as a source or sink of atmospheric $CO_2$ in southern taiga of West Russia.

**Conclusions**

Six years of the paired eddy covariance $CO_2$ flux measurements in 2015-2020 period showed that paludified spruce forest and adjacent ombrotrophic bog in southern taiga of west Russia have a different daily, growing season and annual NEE, TER and GPP as well as different response of the $CO_2$ fluxes to changes in environmental conditions. In spite of the higher daily, growing season and annual TER and



GPP rates at PF site, OB was a stronger sink of the atmospheric $CO_2$ excepting the warmest and the
wettest year of the period (2020) when the growing season was followed by anomalously warm winter
with sparse snow cover. Annual NEE at PF site in the period of measurements ranged between -62 and
145 $gC \cdot m^{-2} \cdot yr^{-1}$, TER between 1366 and 1652 $gC \cdot m^{-2} \cdot yr^{-1}$ and GPP between 1345 and 1592 $gC \cdot m^{-2} \cdot yr^{-1}$.
At OB site annual sums of $CO_2$ fluxes were obtained only for 2020: NEE was -95 $gC \cdot m^{-2} \cdot yr^{-1}$, TER was
410 $gC \cdot m^{-2} \cdot yr^{-1}$ and GPP was 505 $gC \cdot m^{-2} \cdot yr^{-1}$. Cumulative NEE calculated for the long-term growing
season (12 Apr.-11 Oct.) indicated that both ecosystems were predominantly the sink of atmospheric $CO_2$
in the growing seasons of the selected years: NEE varied between -142 and 28 $gC \cdot m^{-2}$ at PF site and
between -132 and -108 $gC \cdot m^{-2}$ at OB site. The lowest annual and growing season NEE was detected in
the warmest year of the period (2020).

Analysis of the seasonal changes in $CO_2$ fluxes and meteorological parameters showed that strong positive
air temperature anomaly in winter can decrease the proportion between growing season and annual GPP
and TER sums especially at PF site. Warm winter with sparse and thin snow cover lead to the increasing
in GPP at PF site more than TER of the paludified forest as well as TER and GPP at the bog. Moreover
warm winter lead to the shifting of the compensation point in spring to the early dates at PF site.

TER at the sites was strongly depended on air and peat temperatures and the GPP was strongly depended
on Rg. Under the anomalously warm weather conditions in late winter and early spring of 2020 the
increased sensitivity of GPP on Rg at PF site was detected. Annual, growing season and daily NEE, TER
and GPP at PF and OB sites as well as its sensitivity parameters to environmental variables are
corresponded to the numerous estimates obtained in the other experimental studies at northern bogs and
forested peatlands. Comparison of the annual NEE at PF site measured in the 2015-2020 period with the
estimates obtained in 1999-2004 period at the same ecosystem showed that PF site switched from the
strong $CO_2$ source to the slight $CO_2$ source and sink close to the $CO_2$-neutral. Changing in $CO_2$ status of
PF site was observed under the positive trends in air temperature, precipitation and ground water level.

We expect that continue of the current warming trend in winter and early spring observed in the region
can increase the $CO_2$ uptake of the peatlands in southern taiga of west Russia especially of the paludified
forests due to the enhanced spring GPP, although the predicted increasing of the drought frequency in
summer can lead to the significant increasing in TER which is able to switch the peatlands into consistent

$CO_2$ source for the atmosphere. Therefore the future $CO_2$ status of the southern taiga peatlands is still unclear. We think that analysis of the interannual flux variability based on the long-term measurements in the peatlands of the different types, management and locations is useful to assess the possible impact

of the weather anomalies, extreme events and climate change on the peatland-atmosphere interaction and can shed light on the future of the northern peatlands.

**Data availability**

The data used in this publication will be provided upon request. The eddy covariance and meteorological data obtained at PF site are available at European Fluxes Database Cluster database (http://www.europe-fluxdata.eu/).

**Author contribution**

VM designed the study, performed the data analysis as well as field measurements at PF site and wrote

the major part of the text. VA designed and performed the field measurements at OB site. DI performed the field measurements and data processing. AV designed the study and field measurements at PF site. JK designed the study and wrote the text.

**Competing interests**

The authors declare that they have no conflict of interest.


 **Aknowledgements**





This study was supported by the grants of the Russian Science Foundation (21-14-00209) and the Russian Foundation for Basic Research (project 19-04-01234-a). The flux data processing and data analysis performed by Mamkin V., Ivanov D., Varlagin A. and Kurbatova J. were supported by the grant of the Russian Science Foundation (21-14-00209). The field measurements provided by Mamkin V. and Varlagin A. was supported by the grant of the Russian Foundation for Basic Research (project 19-04-01234-a).

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
