# Peer review of "Interannual variability of the ecosystem CO2 fluxes at paludified spruce forest and ombrotrophic bog in southern taiga"

_Atmospheric Chemistry and Physics, 2021_

## Author Comment (AC1)

**Response to the Anonymous Referee #1.**

We thank very much Anonymous Referee #1 for the helpful and constructive comments and recommendations. The manuscript has been revised in accordance with reviewer's comments and suggestions to produce an improved version of the article.

**[comment 1]**

*The manuscript compares net ecosystem $CO_2$ exchange from a paludified spruce forest and an adjacent ombrotrophic bog in west Russia and analysed the main environmental controls on NEE and its component fluxes. The study addresses an important research question aiming at better understanding interannual variability in NEE in these understudied ecosystems. The manuscript is mainly well written but could benefit from writing improvements (e.g., grammar and wording). The methodology is sound and in general appropriate for this study. Overall, the study remains very descriptive, and, in my opinion, results should be strengthened by adding uncertainty estimates for fluxes and statistical test results when comparing bog and forest fluxes throughout the manuscript. Additionally, water availability and drought effects are discussed but observational evidence does not appear to support that these factors play a significant role. I think this could be further clarified.*

**Response**

We asked Copernicus English copy-editing service if we need to correct English before the submissions, they replied us that they usually recommend submitting the paper as is, and that English copy-editing is a standard procedure for final revised papers accepted for final publication in ACP.

Uncertainty estimates and statistical analysis has been added to the revised version of the manuscript.

Our research is focused on the $CO_2$ fluxes in 2015-2020 period. The droughts were not observed during this period but such events can potentially influence $CO_2$ exchange of the peatlands in the future. We decided to include only discussion of this effects with corresponding link on the paper where influence of drought on $CO_2$ fluxes was described (Kurbatova et al., 2013).

Kurbatova, J., Tatarinov, F., Molchanov, A., Varlagin, A., Avilov, V., Kozlov, D., Ivanov, D. and Valentini, R.: Partitioning of ecosystem respiration in a paludified shallow-peat spruce forest in the southern taiga of European Russia, Environ Res Lett, 8(4), 045028, doi: 10.1088/1748-9326/8/4/045028, 2013.

**[comment 2]**

*Line 47: The following reference could be relevant here too: Helbig et al., 2019;*

*https://doi.org/10.1029/2019JG005090*

**Response**

We've cited the reference and added it to the reference list (P.2, L. 42-46): "It is suggested that growing air and peat temperatures especially under raising frequency of droughts in boreal regions can significantly increase decomposition rates and switch peatlands from $CO_2$ sink to $CO_2$ source for the atmosphere (e.g. Alm et al., 1999; Moore, 2002; Lund et al., 2012; LaFleur et al., 2015; Helbig et al., 2019; Loisel et al., 2021)."

**[comment 3]**
*Line 54: The following reference could be cited here too: Moore et al., 2006;*
*https://doi.org/10.1111/j.1365-2486.2006.01247.x*

**Response**

We've cited the reference and added it to the reference list (P. 2-3, L. 53-56):"Previous studies reported that NEE of the peatlands is susceptible to water table depth (WTD) dynamics, air and peat temperature variations, changes in global radiation, timing of the snowmelting and peat layer thaw (Moore et al., 2006; Dunn et al., 2007; Lindroth et al., 2007; Sulman et al., 2010)."

**[comment 4]**
*Line 61-64: Helbig et al. (2019) might be relevant here too*

**Response**

We've cited the reference (P. 3, L. 63-66): "For instance, the forest peatlands can sequestrate atmospheric $CO_2$ before the snowmelting and peat thaw in spring, while a thawing is necessary for the beginning of the $CO_2$ uptake at non-forest peatlands (Tanja et al., 2003; Euskirchen et al., 2014; Helbig et al., 2019)."

**[comment 5]**
*Line 72: It might be insightful to include results from the SPRUCE experiment to the*
*introduction and/or discussion (https://mnspruce.ornl.gov)*

**Response**

We've added a several results from SPRUCE experiment to the discussion (P. 31-32, L. 609-616): "While warming and moistening in winter could increase GPP more than TER especially

at paludified forests, the extreme hot and dry conditions are able to increase heterotrophic respiration significantly and switch peatlands from $CO_2$ sink to a consistent $CO_2$ source for the atmosphere as well as alter NPP of the ecosystems. For example, SPRUCE experiment in Minnesota (USA) showed a significant carbon loss rates at black spruce stands on the bog (higher than its historical accumulation rates) under the warming treatment which was connected with increased heterotrophic respiration, decreased *Sphagnum* and tree above ground NPP (Walker et al., 2017; Hanson et al., 2020)."

**[comment 6]**

*Line 78: Another paired flux tower study comparing forested and non-forested peatlands*

*in the sporadic permafrost zone is published by Helbig et al. (2017;*

*https://doi.org/10.1111/gcb.13638)*

**Response**

We've cited the reference (P. 3, L. 75-80): "Unfortunately, in spite of a numerous experimental studies focused on ecosystem-atmosphere $CO_2$ fluxes in different peatland types in high-latitudes in North America (e.g. Roulet et al., 2007; Gill et al., 2017), Europe (e.g. Kurbatova et al., 2002; Lindroth et al., 2007; Minkkinen et al., 2018) and Asia (e.g. Tchebakova et al., 2015; Alekseychik et al., 2017; Park et al., 2021) there is lack of studies considering the ecosystem $CO_2$ fluxes at the forest and non-forest peatlands located in the same landscape and undergo the similar weather conditions (e.g. Euskirchen et al., 2014; Helbig et al., 2017; Zagirova et al., 2019)."

**[comment 7]**

*Line 83: Park et al (2021; https://doi.org/10.3390/atmos12080984) is another study on*

*Russian peatlands.*

**Response**

We have't cited this paper here because the experimental data used in the research was obtained in Asian part of the country.

**[comment 8]**

*Line 93: I think the latitude/longitude coordinates should be listed here for both sites.*

**Response**

We listed the coordinates of the stations to (P. 4, L. 96-99): "This study was conducted at paludified spruce forest (56.4615°N, 32.9221°E, 265 m a.s.l.) and adjacent ombrotrophic bog (56.4727° N, 33.0413° E, 240 m a.s.l.) located on the territory of the Central-Forest state natural biosphere reserve (CFSNBR) in the south-western part of Valdai hills in Tver region of Russia (Fig.1a)"

**[comment 9]**

*Line 106: The growing season definition could already be introduced here.*

**Response**

We've introduced the growing season definition to (P.6, L. 109-114): "Soil surface is typically covered by snow from mid-November to late March - early April (Desherevskaya et al., 2010) and the growing season calculated as the number of days between the first 5-day period with mean daily air temperatures above 5°C (Leaf-on day) to the first 5-day period with mean daily air temperatures below 5°C (Leaf off day) following (Urban SIS, 2018; Buitenwerf et al, 2015; Donat et al, 2013; Mueller et al, 2015) lasts 182 days on average (since 12 Apr. to 11 Oct.)."

**[comment 10]**

*Line 146: It is unclear what a "standard design" is. Please clarify.*

**Response**

We've deleted the word "design" from the (P.7, L. 155): "Flux stations at PF and OB sites have a standard instrumentation for FLUXNET network."

**[comment 11]**

*Line 148: The tower height is 29 m, but trees reach up to 27 m. It seems as if the EC measurements could be most of the time in the roughness sublayer affecting the validity of the essential EC assumptions. Perhaps the authors could explain how this potential issue was addressed.*

**Response**

We've updated the information about tree heights and also, added DBH. According to the last survey at PF site (November 2021) the mean tree height around the tower is 16.9 m, but several trees in the ecosystem reaches 27 m (P. 6, L. 130-131): "The mean tree height is 16.9 ±6.4 m (±SD) with mean diameter at breast height (DBH) 21.6±8.9 cm (±SD) and undergrowth is about 0.3 m.".

**[comment 12]**

*Line 200: Why was VPD not included in the GPP response?*

**Response**

We haven't included VPD in the GPP-Rg response analysis due to the lack of GPP data obtained under high VPD. VPD at the ecosystems changed in the narrow range: in early spring and late autumn day-time VPD at PF and SB sites didn't exceed 5 hPa, even in the driest summer 2018 about 90% of the 30-min day-time VPD values were less than 10 hPa and about 75% of the day-time data were less than 5 hPa.

**[comment 13]**

*Table 1 and other tables: At least for the long-term means, the standard deviation*

*should be included in the table. It would also help to characterise how strong the climate*

*anomalies were.*

**Response**

We've added the standard deviations of the long-term mean air temperature and precipitation values to Table 2 and Table 3.

**[comment 14]**

*Line 262: Is there a relationship between precipitation and water table depth?*

**Response**

As it shown in Tab. 1 mean annual WTD negatively correlated with annual precipitation, but parametrization the relationship between precipitation and water table depth on shorter scales is challenging due to the lagged response of the WTD on precipitation.

**[comment 15]**

*Line 273: This is one example where the claim that "strong dependence ... was not found"*

*should be backed up with statistical methods.*

**Response**

We've decided not to consider the dependence between the annual sums of the $CO_2$ fluxes and GSL due to the small number of samples. We expressed the sentence in another way (P. 14, L.290-292): "Maximal and minimal annual NEE, TER and GPP at PF site were not correspondent with maximal and minimal GSL."

**[comment 16]**

*Line 284: It is unclear where this hypothesis is coming from and how it is backed up.*

**Response**

We've removed this sentence from the text.

**[comment 17]**

*Table 3 and other flux tables: Should include uncertainties in aggregated fluxes.*

**Response**

We've added uncertainties to Table 3 and Table 4.

**[comment 18]**

*Line 324: Leaf-on and leaf-off might not be the right terms for evergreen ecosystems.*
*Start and end of growing season might be more accurate.*

**Response**

We've replaced the words "leaf on" and "leaf off" with "the start" and "the end" of the growing season everywhere in the text.

**[comment 19]**

*Line 355: It seems as if the Q10 model was fitted to the entire dataset. Did the authors consider analysing short-term variations in temperature sensitivity (see Reichstein et al., 2005; https://doi.org/10.1111/j.1365-2486.2005.001002.x).*

**Response**

We've considered the short-term variations of the temperature sensitivity and tried to find the difference between the years, months, periods and WTD classes, but these differences were very small. Thus, we've decided to include only best fitted models to show the difference between the sites. We also found it challenging to demostrate the difference between $Q_{10}$ models fitted for short-term periods due to the lack of original nighttime NEE data obtained at the sites.

**[comment 20]**

*Conclusions: In my opinion, the conclusion would be more impactful if it was shortened and if the main take-home messages were highlighted here.*

**Response**

We've added the new version of conclusions to the text.

---

## Author Comment (AC2)

**Response to the Anonymous Referee #2.**

We thank very much Anonymous Referee #2 for the helpful and constructive comments and recommendations. The manuscript has been revised in accordance with reviewer's comments and suggestions to produce an improved version of the article.

**[comment 1]**

*In "Interannual variability of the ecosystem CO2 fluxes at paludified spruce forest and ombrotrophic bog in southern taiga", Mamkin et al. present CO2 flux data and analysis at two taiga peatland sites in western Russia. They highlight the interannual variability in the CO2 fluxes and driving meteorological and environmental conditions at and between both sites, with implications for the future net carbon balance of this region and ecosystem due to climate change.*

*Overall, this is an important topic and the study presented here is, for the most part, thoroughly and completely introduced, described, and discussed, with results placed in a proper context. The study data are great for long term climate trends in a sparsely monitored region, and the paper shows well how ecosystem warming has varied impacts depending on seasonal timing. However, many English-language errors greatly hinder the paper's readability and must be corrected. Additionally, all discussion of uncertainty in the CO2 flux measurements and partitioning methods is absent and must be included prior to publication in ACP.*

**Response**

We asked Copernicus English copy-editing service if we need to correct English before the submissions, they replied us that they usually recommend submitting the paper as is, and that English copy-editing is a standard procedure for final revised papers accepted for final publication in ACP.

Uncertainty estimates and its discussion have been added to the revised version of the manuscript.

**[comment 2]**

*More specific comments and suggestions are listed below:*

*Figure 1a: This figure would be more useful with country borders, lat/lon descriptions, and more contrast in colors between the different land cover types.*

**Response**

We've changed the Figure 1a.

**[comment 3]**

*Line 100: I wondered why air temperature was not used from MS site when introduced here. It is later mentioned to be not available, perhaps move this mention earlier.*

**Response**

We've added this information to (P. 5-6, L. 106-109): "Long-term mean annual precipitation (1991-2020) measured at meteorological station "Lesnoy Zapovednik" (56.50° N, 32.83° E, 240 m a.s.l.) – the nearest meteorological station to the study area was 778 mm (continuous air temperature data from "Lesnoy Zapovednik" meteostation for 1991-2020 period is not available)".

**[comment 4]**

*Line 108: Add additional context for CMI range of values, for those not familiar.*

**Response**

We've added information about CMI index to (P. 6, L.115-118): "The climate moisture index (CMI) calculated as the ratio of annual precipitation to annual potential evapotranspiration and ranged between -1 and 1 (Wilmott and Feddema, 1992) is 0.3 - 0.4 (Mamkin et al., 2019; Novenko et al., 2015)."

**[comment 5]**

*Line 110: At which site or sites is this regional trend detected?*

**Response**

We've corrected the sentence (P. 6, L.118-120): "In the recent 30 years a positive trend of air temperature (+0.73 °C per10 years) and precipitation (+3.6 mm·month-1 per 10 years) was detected at the meteorological stations "Toropets" and "Lesnoy Zapovednik" respectively".

**[comment 6]**

*Figure 1b: Not cited in text.*

**Response**

We've added the citation to (P. 4, L.99-100): "The sites are located 7.5 km apart (Fig. 1b) and characterized by very similar weather conditions."

**[comment 7]**

*Line 165: This paragraph continues description of OB site, but the paragraph break without further mention of OB makes this unclear.*

**Response**

We've added the site name to the first sentence of the paragraph (P. 8, L. 174): "Additionally, global radiation at OB site was measured with 4-component radiometer NR01 (Hukseflux Thermal Sensors, The Netherlands) at 2.5 m height."

**[comment 8]**
*Line 176: Should this be "2015-2020"?*
**Response**

We've corrected the sentence (P. 8,L.185-186): This study is based on eddy covariance and meteorological data obtained at PF and OB sites in 2015-2020.

**[comment 9]**
*Line 179: What about the "2" quality flag makes that flux worthy of being removed?*
**Response**

We edited the sentence (P. 8, L.189-190): "All fluxes with quality flag 2 was removed from the analysis following the recommendations on the data quality assessment (Mauder et., 2013)."

**[comment 10]**
*Lines 176-184: As mentioned above, this section must be expanded to include description of error and uncertainty associated with eddy flux measurement, calculation, and partitioning of GPP and TER from observed NEE. Perhaps a comparison of the derived TER and GPP from isolated NEE alone (section 2.4) with the automated partitioning would be useful. Further discussion later on should reference how the results could differ based on the potential errors and uncertainty.*

**Response**

We've added the information about uncertainty estimation to (P. 9, L.196-201): "Uncertainty of NEE, TER and GPP associated with the random error in the measured fluxes, u*-threshold estimation, gap-filling and flux partitioning procedures was calculated using REddyProc package (Wutzler et al., 2018) as standard deviation (SD) of the flux values. The aggregated random uncertainty of the seasonal and annual sums of the $CO_2$ fluxes was obtained considering the autocorrelation between the residuals using empirical autocorrelation function (Zięba & Ramza,

2011)." Also, a section considering the flux uncertainties have been added to the discussion (P. 29, L.537-548).

**[comment 11]**

*Figures 2 and 3: It may be more effective to convey interannual variability in meteorology and CO2 flux as anomalies from a mean set of values, rather than a timeseries. This is especially true when referring to differences on a monthly scale, such as early snow-off in a particular year.*

**Response**

We absolutely agree that the chart with anomalies would be more useful than a timeseries graph in this context. However, we've decided to use a timeseries because of several long gaps in the data obtained at OB site that makes difficult to calculate flux and meteorology anomalies comparable with the anomalies derived for PF site. At least, timeseries can show to readers the general differences in seasonal dynamics of the $CO_2$ fluxes between two peatlands in spite of long gaps.

**[comment 12]**

*Line 269 and elsewhere: Considering add in mention of processes when referring to numbers such as NEE. Rather that or in addition to saying "NEE decreases", one could say "net CO2 uptake increases".*

**Response**

We've corrected the sentence at (P. 14, L.283-284) and the similar sentences throughout the manuscript: "During the 6 years of measurements $CO_2$ uptake at PF site tended to increase."

**[comment 13]**

*Line 370: Why does GPP determine the parameters between the sites? Because of relative constant Rg?*

**Response**

The difference of the parameters between the sites was mostly determined by GPP due to the almost equal Rg (difference of the daily sums between the sites was on average ±3%).

**[comment 14]**

*Lines 373-379: Was there similar (any?) interannual variability in the TER parameters as for GPP shown here?*

**Response**

The analysis of the interannual variability in the TER parameters hasn't been included to the text due to the small differences between them. We also found it challenging to research the difference between $Q_{10}$ models fitted for short-term periods due to the lack of original nighttime NEE data obtained at the sites.

**[comment 15]**

*Line 473: The predictive relationships between environmental drivers and CO2 fluxes mentioned here are not shown. A figure or statistics that illustrate these would be useful*

**Response**

As an example, we've added the relationship between the residuals of the $Q_{10}$ models (calculated using soil temperature) and WTD (Figure 5).